# When Research Evidence and Healthcare Policy Collide: Synergising Results and Policy into BRIGHTLIGHT Guidance to Improve Coordinated Care for Adolescents and Young Adults with Cancer

**DOI:** 10.3390/healthcare13151821

**Published:** 2025-07-26

**Authors:** Rachel M. Taylor, Alexandra Pollitt, Gabriel Lawson, Ross Pow, Rachael Hough, Louise Soanes, Amy Riley, Maria Lawal, Lorna A. Fern

**Affiliations:** 1Department of Targeted Intervention, University College London, London WC1E 6BT, UK; 2Centre for Nurse, Midwife and Allied Health Profession Led Research (CNMAR), University College London Hospitals NHS Foundation Trust, London NW1 2PG, UK; 3Policy Institute, King’s College London, London WC2B 6LE, UK; alexandra.pollitt@kcl.ac.uk (A.P.); gabriel.lawson@kcl.ac.uk (G.L.); 4Power of Numbers Ltd., Cambridge CB22 6SJ, UK; ross.pow@powerofnumbers.co.uk; 5Teenage and Young Adult Cancer Service, University College London Hospitals NHS Foundation Trust, London NW1 2BU, UK; rachaelhough@nhs.net; 6Cancer Institute, University College London, London WC1E 6DD, UK; 7Teenage Cancer Trust, London WC1V 7AA, UK; louise.soanes@teenagecancertrust.org; 8Patient Representatives; 9Cancer Clinical Trials Unit, University College London Hospitals NHS Foundation Trust, London NW1 2PG, UK; lorna.fern@nhs.net

**Keywords:** adolescent, teenager, young adult, BRIGHTLIGHT, service delivery, model of care, healthcare policy

## Abstract

**Background/Objectives:** BRIGHTLIGHT was the national evaluation of adolescent and young adult (AYA) cancer services in England. BRIGHTLIGHT results were not available when the most recent healthcare policy (NHSE service specifications for AYA Cancer) for AYA was drafted and therefore did not consider BRIGHTLIGHT findings and recommendations. We describe the co-development and delivery of a Policy Lab to expedite the implementation of the new service specification in the context of BRIGHTLIGHT results, examining the roles of multi-stakeholders to ensure service delivery is optimised to benefit AYA patients. We address the key question, “What is the roadmap for empowering different stakeholders to shape how the AYA service specifications are implemented?”. **Methods:** A 1-day face-to-face policy lab was facilitated, utilising a unique, user-centric engagement approach by bringing diverse AYA stakeholders together to co-design strategies to translate BRIGHTLIGHT evidence into policy and impact. This was accompanied by an online workshop and prioritisation survey, individual interviews, and an AYA patient workshop. Workshop outputs were analysed thematically and survey data quantitatively. **Results:** Eighteen professionals and five AYAs attended the face-to-face Policy Lab, 16 surveys were completed, 13 attended the online workshop, three professionals were interviewed, and three AYAs attended the patient workshop. The Policy Lab generated eight national and six local recommendations, which were prioritised into three national priorities: 1. Launching the service specification supported by compelling communication; 2. Harnessing the ideas of young people; and 3. Evaluation of AYA patient outcomes/experiences and establishing a national dashboard of AYA cancer network performance. An animation was created by AYAs to inform local hospitals what matters to them most in the service specification. **Conclusions:** Policy and research evidence are not always aligned, so when emerging evidence does not support current guidance, further exploration is required. We have shown through multi-stakeholder involvement including young people that it was possible to gain a different interpretation based on current knowledge and context. This additional insight enabled practical recommendations to be identified to support the implementation of the service specification.

## 1. Introduction

BRIGHTLIGHT was the national evaluation of adolescent and young adult (AYA) cancer services in England conducted between 2012 and 2019 [1]. This paper reports the process of linking research evidence to clinical practice and policy recommendations.

### 1.1. Delivery of Healthcare in the United Kingdom

Healthcare in the United Kingdom (UK) is free at the point of access and based on clinical need, regardless of the ability of the patient to pay. In England, secondary care services are delivered in National Health Service (NHS) Trusts, which consist of one or more hospitals in a specific geographical region. While funding for the NHS is centralised through the Department of Health and Social Care, each NHS Trust functions independently, delivering care based on a multitude of factors, such as ensuring equity in access to care, national policy guidance, and budget allocation [2]. Clinical Commissioning Groups (CCGs) were established in 2012 in England to ensure services delivered in secondary care fulfilled local care needs. The CCGs were led by a clinician and had representation of a range of key stakeholders working with patients and the wider clinical team in the local community to ensure the services they commission address local needs [3].

In addition to the commissioning of services through the CCGs, Clinical Reference Groups were established to “provide expert clinical advice and leadership relating to specialised services” [4]. Specialised commissioning includes treatments for rare and complex conditions to ensure funding is provided in addition to standard care, so existing services are supported and continue as usual, but more complex care is also provided for. Specialised commissioning is based on best available evidence and involves a level of centralisation, i.e., not every NHS Trust in England provides specialised services, only those who fulfil the criteria developed by the Clinical Reference Group [4]. Four aspects of cancer care that are covered through specialised commissioning are radiotherapy, chemotherapy (and other anti-systemic cancer treatment), specialised cancer surgery, and children and young people’s cancer services [5].

### 1.2. Delivery of Cancer Services for Young People in England

Cancer care in England for AYAs aged 16–24 is a speciality distinct from children and adult cancer services in recognition that they have more complex needs that extend beyond standard cancer care and treatment [6,7]. While cancer in this age group is uncommon, with approximately 1% of all new diagnoses in the UK, survival outcomes are poorer in comparison to some children and older adult cancers [8]. Over the past three decades, specialised cancer services for AYAs in the UK have evolved considerably, related to the active involvement of the charitable organisation Teenage Cancer Trust [9,10], who opened the first adolescent cancer unit in 1990, followed by formal recognition of unique care needs by the release of the National Institute for Health and Clinical Excellence (NICE) Improving Outcome Guidance for Children and Young People in 2005. This document led to centralising AYA cancer services in England to 13 Principal Treatment Centres (PTCs) and provided a roadmap for AYA services, covering clinical organisation, facilities, diagnostic tools, and treatment options, including access to clinical trials. However, recommendations were mainly drawn on evidence from children’s and adult studies with only a limited amount of evidence specifically focusing on young people [11]. Despite this national document, national disparities persisted around where and how young people received their care [12].

By 2012, the model for delivering cancer care to young people in England was organised into 13 AYA networks of care. Each of these networks consisted of a PTC, which offered expertise in treating the range of cancers prevalent among young people. These PTCs were also supported by dedicated AYA multidisciplinary teams (MDTs) focused on addressing the psychosocial needs of this patient population within an environment tailored to the developmental and social requirements of young people [9]. For adolescents up to the age of 16, the option was to receive care in a children’s specialist cancer unit or a paediatric oncology shared care unit, which could provide specific aspects of supportive care, such as the administration of blood products or simple chemotherapy drugs. Adolescents aged 16 to 18 needed to be treated in the PTC, while young people aged 19–24 were to have ‘unhindered access’ to age-appropriate care, and the choice of either being referred to a PTC or staying in an adult cancer unit in a designated hospital within the AYA network.

Designated hospitals were self-nominated to provide core aspects of cancer care to achieve the standards agreed by the Clinical Reference Group [13]. They were required to notify the PTC of newly diagnosed young people to ensure a ‘sharing of responsibility for patient management’ between the clinical team specialising in the tumour site at the designated hospital and the experts at the PTC. Furthermore, young people at designated hospitals were expected to have ‘unhindered access’ to the support of the broader MDT through outreach efforts by specialist professionals from the PTC, for example, young people’s social workers [11]. Within each network, there were also hospitals not allocated to provide care to young people (non-designated hospitals). Some continued to deliver care in these non-designated hospitals, lacking access to the age-specific expertise of the AYA MDT at the PTC; therefore, young people missed out on age-appropriate care [14].

Young people had multiple options for receiving cancer care in England, with variations in the translation and implementation of services based on factors such as disease type, age, location, and service availability [15]. Notably, in 2013, among the 76 hospitals designated to deliver AYA cancer care, approximately one-third were unable to meet 50% or more of the specified standards for designation [13]. There were no repercussions for these hospitals, and they remained designated for AYA care, despite lacking many essential elements of a young-adult-friendly cancer service [16].

### 1.3. National Evaluation of AYA Cancer Services

In 2010, the guidance and policy directing cancer services for young people in England lacked an evidence-based foundation, and there was a notable absence of research assessing the effectiveness of the PTC and associated networks. BRIGHTLIGHT, an applied health research program, emerged from the Essence of Care study’s feasibility work [17,18,19,20], which shaped its methodology. The Essence of Care study revealed significant variations in care delivery across England, indicating the need for any evaluation to be multicentre and contain a longitudinal element. BRIGHTLIGHT was launched in 2012 to answer the question: Do specialist cancer services for AYA add value? This programme of work contained six distinct but interlinked studies to explore the competencies of healthcare professionals providing care [21,22], the environment in which care was delivered [15,23], the experiences and outcomes of young people receiving care [24,25,26] and their caregivers’ experiences [27], and the cost of cancer care to young people and to the NHS [1].

Central to the BRIGHTLIGHT programme was a nationwide cohort study of young people aged 13–24 at diagnosis and followed longitudinally for 3 years [28]. There were significant multifactorial challenges in cohort recruitment, including evolving regulatory processes, which led to the programme of research taking longer than anticipated. This was unfortunate, as in 2019, the children and young people’s cancer Clinical Reference Group was tasked with reviewing the evidence to draft an updated service specification for the PTC and designated hospitals [29]. BRIGHTLIGHT evidence was not available to contribute to this. The subsequent cohort results indicated that specialist AYA cancer care in 2012–2014 was not associated with better quality of life or survival, and young people who received care in both the PTC and either a children’s or adult cancer centre had the poorest outcomes [24,26,30]. In contradiction to the BRIGHTLIGHT cohort findings, a key change proposed in the draft service specification was for ‘Joint care’ between PTCs and designated hospitals, now formally named Operational Delivery Networks (ODNs), meaning that the revised guidance advocated for care that had previously been shown to be associated with poorer outcomes than for those receiving all their care in one hospital [29]. Further details of BRIGHTLIGHT results are shown in Table 1 and Table 2 and Figure 1 and the accompanying text in ‘Section Summary of the Evidence Used in the Policy Lab’.

While BRIGHTLIGHT cohort results were in conflict with this proposed change, the qualitative element of the BRIGHTLIGHT programme afforded a more nuanced understanding of the way cancer services were delivered and the potential benefits [23]. In 2012, when young people were recruited to the cohort, there were a few PTCs who were well established within their geographical region, but most were in their infancy and were developing relationships with the designated hospitals and implementing the NICE guidance. The case study was conducted in 2015 when the PTCs were becoming more embedded in their networks, and one of the main findings was that time was needed for a culture of AYA cancer care to be established across a network [23]. We could surmise, therefore, that as the ODNs mature, the coordination of care would be better, and consequently, young people will have improved experience and outcomes compared to AYAs receiving care between hospitals in 2012. This is being explored in a subsequent study [31].

By November 2022, the service specification remained in draft, with an imminent release. However, it was assumed that the recommendation for ‘joint care’ would remain despite the BRIGHTLIGHT cohort results. We set out to co-develop and deliver a Policy Lab to expedite the implementation of the new service specification in the context of the BRIGHTLIGHT results, examining the roles of multi-stakeholders to ensure service delivery is optimised to benefit AYA patients. We address the following key question:

“What is the roadmap for empowering different stakeholders to shape how the AYA service specifications are implemented?”

## 2. Methods

This project used a method developed by the Policy Institute at King’s College London: a Policy Lab, which is a process for “facilitating research evidence uptake into policy and practice” [32]. This is a collaborative process engaging a wide range of stakeholders, including patients, and fitted with the co-production model employed in the BRIGHTLIGHT study, which had significant patient involvement [33]. The stages of a Policy Lab are summarised in Box 1.

Box 1The stages in organising a Policy Lab [34].Stage 1: Set aside time for planningStage 2: Establish the need and purpose of the Policy LabStage 3: Select and invite participantsStage 4: Synthesise and communicate evidenceStage 5: Plan agenda and facilitationStage 6: Conduct the Policy LabStage 7: Report the resultsStage 8: Create and support the new coalition

### 2.1. Stages 1 and 2: Planning and Establishing the Purpose of the Policy Lab

There are multiple purposes of a Policy Lab, spanning from issue identification to policy evaluation [32]. The policy (commissioning specification) for AYA cancer care, albeit in draft, had already been created; therefore, the aim of this Policy Lab was to focus on its implementation. The Policy Institute and BRIGHTLIGHT research teams collaborated with the chair of the children and young person’s Clinical Reference Group and the Chief Nurse at Teenage Cancer Trust (who fund much of the infrastructure and workforce for AYA cancer care in the UK) to develop the question to be addressed in the Policy Lab. It was agreed that the focus should be on the ODN to align with BRIGHTLIGHT evidence, e.g., How do we ensure that care is coordinated across the ODN in a joint care model? The questions were felt to be important, as meaningful engagement with stakeholders was viewed as important to ensure coordinated care. The proposed question was, therefore, the following:


*“What is a roadmap for empowering different stakeholders to shape how the teenage and young adult service specifications are implemented?”*


### 2.2. Stage 3: Select and Invite Participants

We aimed to include the key stakeholders who were going to be impacted by the new service specification or were responsible for implementing it, as they would have firsthand experience with current practice and could offer valuable insights into both barriers and facilitators to recommendations. Participant eligibility included all stakeholders involved in the commissioning, provision of, and delivery of clinical and psychosocial care to AYAs with cancer. This included commissioners, third sector (not-for-profit, non-government-funded organisations), clinical leads (nurses and doctors), ODN managers, representation from national research bodies, and, importantly, young people with experience of cancer. Invites were sent directly to the lead commissioner for Children and Young People’s Cancer, chair of the National Cancer Research Institute (NCRI) Teenage and Young Adult Research Group, and the children and young person’s cancer clinical representative at the National Institute for Health and Care Research (NIHR). The commissioner was asked to circulate the invite to the managers and clinical leads of the ODNs. The ODNs were asked to send one representative from each PTC, as capacity for the Policy Lab was limited. Exclusion criteria for professional stakeholders were those not involved in the AYA cancer. Patient representatives were recruited from the BRIGHTLIGHT Young Advisory Panel (YAP). AYAs are eligible to participate in the YAP if they have been diagnosed with any cancer between the ages of 13 and 24 years and diagnosed within the last 10 years (in keeping with the BRIGHTLIGHT recruitment period, 2012–2014). Exclusion criteria for the YAP are those with a primary cancer diagnosis aged less than 13 years and older than 25. The YAP had been involved with BRIGHTLIGHT since the study began and had worked alongside the research team in study development and management [33]. The group comprised 10 young people aged 13–24 years at the time of diagnosis, currently aged 26–29 years. An invite was sent through their secure Facebook page.

The Policy Lab was planned as a whole-day, in-person workshop in central London. To ensure equity of access, travel was provided, refreshments were provided throughout the day, and the members of the YAP received payments in line with national guidance [35]. No regulatory approvals were required for the Policy Lab, which was classed as a dissemination activity, but the ethical processes approved as part of the BRIGHTLIGHT cohort were applied (HRA references: 11/LO/1718 and ECC 8-05(d)/2011).

### 2.3. Stage 4: Synthesise and Communicate the Evidence

All participants were sent a ‘briefing pack’ one week before the Policy Lab to ensure that they were equipped with sufficient knowledge of the evidence to be comfortable engaging in the day’s activities. The research team identified the key messages from the final NIHR report, which contained a synopsis of the six BRIGHTLIGHT studies [1]. The Policy Institute team translated these into an accessible summary document in a format that could be understood by all the participants, with links to the full report and academic publications should they want more information (Appendix A).

#### Summary of the Evidence Used in the Policy Lab

Young people’s experience of care: A total of 830 young people comprised the BRIGHTLIGHT cohort [28]. A summary of the key cohort findings is presented in Table 1 [24,26,30].

The workforce delivering care: The top five healthcare professional competencies identified from the international Delphi survey as part of the BRIGHTLIGHT programme are presented in Table 2 [26], highlighting the distinct skill set needed in order to deliver high-quality AYA care. Competence was required not just in cancer-related care but also in young person-related care.

The environment in which care is delivered: A definition of age-appropriate care was developed from existing evidence and analysis of interviews conducted in the case study. ‘Age-appropriate care’ was a complex term that was explained through a conceptual model showing that care delivered in an environment that promoted normality was essential to the delivery of optimal holistic and young person-centred care [36]. The formation of a culture responsive to the unique needs of AYAs was influenced by four factors but required time for it to develop (Figure 1) [23].

### 2.4. Stage 5: Plan Agenda and Facilitation

The research team and the Policy Institute team met several times prior to the Policy Lab to ensure understanding of the key issues and seamless facilitation of the day. The agenda was designed to guide the group through a collaborative process of generating ideas and then focusing on practical recommendations.

### 2.5. Stage 6: The Policy Lab Workshop

The Policy Lab was a whole-day workshop from 10:00 to 16:00, with breaks. The room was organised around four circular tables, each table having a member of the BRIGHTLIGHT research team and at least one young person. The other participants rotated around the tables after each activity to stimulate new discussions. There were four activities that participants worked on in their groups and then fed back to the whole room.

Reviewing the briefing pack: discussion guided by three key questions:What is particularly striking or surprising about the information in the briefing pack?What recent developments or other information do you know of that would be good to share with the rest of the group?What questions would it be helpful to explore during the rest of the day?Bridging the gap between the specification and evidence. Addressing the difference in the outcomes (Table 1) between all, no, and some care in the PTC:Why do you think the outcome is poorer compared to no access to the PTC?What can be done to address this in implementing the service specification?Identifying options for implementing the service specification. The evidence (Section Summary of the Evidence Used in the Policy Lab) was reviewed, and each group was asked to discuss the following: Which things are you most confident in being able to implement?Which things might be harder to implement?What would you prioritise to get stakeholders’ input on?
Developing a future roadmap for stakeholder contributions. Teams were asked to draw their roadmap for empowering stakeholders in the implementation of the service specification, identify how this could be supported, and the metrics that would show success.

Data from the Policy Lab included the templates used on each table for recording the discussions and the notes taken during the whole-group discussion. Following the Lab, this data was analysed through content analysis and written up in the form of a ‘Pyramid Narrative’, which contained the key conclusions and supporting arguments. These conclusions were later used to construct a policy report, in which the recommendations were expounded, alongside data from the follow-up workshop described below. The analysis was conducted by the Policy Lab team and independently checked by the research team against the raw materials.

### 2.6. Follow-Up Workshop

Participants in the Policy Lab expressed interest in holding an additional workshop to develop actions to support the implementation of some of the recommendations. The follow-up workshop was planned as a two-hour virtual meeting on Zoom. There were no restrictions on the number of people who could attend, so the clinical leads and managers of the ODNs were asked to circulate the invitation email to the clinicians across their networks.

Prior to the workshop, a survey was circulated to the Policy Lab participants via an online survey platform (Qualtrics). They were asked to rank the national-level priorities, then consider the following:Other areas where there could be national-level co-operation.Suggestions for implementing the recommendations.Identify any recommendations that would be particularly challenging to implement.Based on the priorities, outline any examples of good practice already happening at a local level.Present any developments since the Policy Lab, which might influence implementation.

The results of the survey formed the structure and content of the workshop. Anyone who was unable to attend the workshop but wanted to contribute had the option of taking part in a semi-structured interview. The workshop focused on the top three priorities identified in the ranking exercise in the survey. In breakout rooms supported by a facilitator, participants discussed the ideas they had to progress the recommendation. These were fed back to the group as a whole, to elicit further discussion. Finally, using Mentimeter (https://www.mentimeter.com/ (accessed on 23 May 2025)), participants anonymously allocated 100 points to the proposals they liked the most, rated the confidence they had that these ideas could progress, and shared any other ideas that had not been expressed.

### 2.7. YAP Workshop

Following the Policy Lab and online workshop, nine members of the YAP were invited to a one-day face-to-face workshop to discuss key changes in the service specification. The discussion was audio recorded, and the transcript was used as the basis for the script for an animation to support informing the DHs of key points of the service specification. This transcript was written by the research team and then reviewed and approved by the YAP. The animation was drafted by an animator experienced in working in healthcare.

### 2.8. Dissemination of Results to Participants

Participants, including the young people, were emailed a final report of the Policy Lab. Results were disseminated through normal academic and professional channels. The YAP animation was sent to the young people.

## 3. Results

Figure 2 illustrates an overview of key steps of the Policy Lab, the number of participants at each stage, and the outputs generated.

### 3.1. Stage 7: Reporting the Results

A total of 26 people participated in the Policy Lab: three members of the research team; five members of the YAP; representatives of the NCRI, NIHR, Clinical Reference Group, NHS England, and the third sector; and representatives from the ODNs included one doctor, seven nurses, and five network managers. The Policy Lab was held in November 2022, prior to the launch of the service specification. The delay in launching the specification was of utmost concern to the clinical leads and network managers. Three themes arose from the discussions during the Policy Lab:Ambitions for service specification implementation.Addressing the potential gaps between the service specification and feedback from the BRIGHTLIGHT research.Roadmap for empowering stakeholders to shape implementation of the AYA cancer service specification.

#### 3.1.1. Ambitions for the Service Specification

Participants discussed their ambitions for the service specification during the policy lab:The implementation should aim for all individuals to receive standardised, high-quality care regardless of where they live and where they are treated.Viewing the delivery of care as a shared endeavour between designated hospitals and the PTCs. Designated hospitals should not be seen as a “step down”, given that some patients will choose these because of a preference for being treated locally. The result of the specification implementation should not result in PTCs “being pitted” against designated hospitals but rather working together “as part of a package”.The importance of patients “feeling like a young person” was especially crucial to the design of care settings and service delivery.The pathway to diagnosis (e.g., through primary care or via the emergency department) could be a cause of significant variation between young people. This was a policy gap that implementation of the service specification should try to pick up.Attention should be paid to looking out for people “falling through the gap” between PTCs and the designated hospitals, e.g., offering to pay expenses for food, travel, and respite care or ensuring access to antibiotics and other necessary support.Involving young people with experience of cancer treatment (e.g., following end of treatment) in the mentoring of other young people could be very valuable in guiding them through their journey, helping them avoid or mitigate trauma (e.g., by accessing relevant support) and identifying questions to ask at different points in the process.Each network should consider having a single person that young people or their caregivers can go to with questions (e.g., keyworker or navigator, to ask what a meeting with clinicians is about, to get help with claiming costs, to ensure their feedback is heard). This is a different, though related, role to that of an AYA clinical nurse specialist and could be funded to work across the network. The person should be a passionate advocate with enough understanding of the issues affecting young people.The specific value of having an AYA clinical nurse specialist should be promoted, and hospitals should be encouraged to work with charities to ensure these are in place, and then the funding for these posts ringfenced or protected as far as possible.Accessing clinical trials on an equitable geographic basis is complex and daunting but is something the networks should focus on strongly.

#### 3.1.2. Addressing the Gap Between the Specification and BRIGHTLIGHT Evidence

The BRIGHTLIGHT research showed that for some metrics, the results were poorer for those receiving care based in PTCs than those who did not. Participants at the Policy Lab explored why this may be the case and pointed to several things that could be borne in mind as part of implementing the service specification.

Illness perception: There was a spread of opinions around how the feedback on illness perception could be interpreted. In broad terms, there was agreement that rather than looking on these as ‘good’ or ‘bad’, it pointed towards there being benefits in young people having exposure to other young people in the PTC, sharing their experiences, and acknowledging that “*you are allowed to feel ill*” or “*can feel like a cancer patient*”. It was felt that this might be helpful with treatment adherence. The route taken along the pathway to get to the PTC (including prolonged routes to diagnosis) would also have an impact on illness perception.

Quality of life 6 months after diagnosis: Young people experienced different routes to diagnosis, for example, whether the initial contacts were within primary care or via emergency departments. Those who took longer to be told (from symptom onset to first cancer consultation/diagnosis) were more likely to be anxious, depressed, and report lower quality of life. It was agreed that differences between services in the age cut-offs at either 16 or 18 caused variations in experience, and there was the risk that young people could fall into gaps in clinical care and access to support services (e.g., community nursing).

Involvement in decisions: Lower involvement in decisions raised a discussion about “*paralysis at the point of diagnosis*” and the need for guidance and support for young people to make decisions. There was agreement that it was important to keep reassessing how much support was needed or wanted, and this should be from initial diagnosis (when it was felt that no one should be given this alone) all the way through to when transitioning back to day-to-day life after completion of treatment.

Participants noted communication and information were key to being involved in decisions. At all stages of the treatment journey, having an advocate for young people was an opportunity for peer interaction to receive practical advice. It was suggested that professionals could explore the use of technology (e.g., QR codes) to access information in an easy and timely way. Input from young people was noted to be valuable to develop messaging and prompts that gave honest answers to the range of typical questions/scenarios that young people experienced during their treatment journey. The development of patient portals for disseminating this information (and ideally also for allowing for the easy access and sharing of patient records), either at the national level or in the local network, would also be highly desirable.

Travel costs and out-of-pocket expenses: Participants were not surprised that the costs of travel were higher to PTCs given the additional length of journeys likely to be involved. Funding provided to young people for travel was the same amount, regardless of the duration or location of the treatment. For some, treatment lasts years, while others have a one-off intervention at the PTC. It was agreed that efforts should be made to make the support for expenses simpler, for example, by not making the payments retrospectively. The question of costs to the patient raised the question about the relative value of treatment: “*what is worth the travel?*”. It was suggested that the design of joint care could include regular check-ups locally, with only something more complex justifying the time and money spent travelling to the PTC. While there may be a willingness to travel, the cost-of-living pressures may mean young people could not readily afford this, and not easily being able to take up PTC-based treatment options might result in additional stress.

Costs to the NHS: Concern was raised that without ring-fenced funding, it might be challenging for ODNs to deliver the new specification as intended. This was particularly the case in places where there was a big variation in the current provision compared with what the specification proposed.

#### 3.1.3. Roadmap for Empowering Stakeholders

Participants in the Policy Lab suggested a set of actions for how ODNs could engage and empower different stakeholders to obtain their support and involvement in implementing the service specification. These were at a national level (Table 3) and more local (Table 4).

### 3.2. Results from the Follow-Up Workshop

Sixteen people from six ODNs completed the survey, which identified the top three national priorities to address in the workshop:Launching the service specification supported by compelling communication.Harnessing the ideas of young people.Evaluation of AYA patient outcomes/experiences and establishing a national dashboard of AYA cancer network performance.

Three people participated in the one-to-one interviews, and 13 people attended out of the 25 who had registered to join the virtual workshop: 7 nurses, 4 ODN managers, 1 doctor, and a member of the YAP. The follow-up workshop was held in June 2023, two weeks after the service specification was officially launched [37].

#### 3.2.1. Launching the Service Specification Supported by Compelling Communication

Comments stressed the importance of robust communication and the ideal timing given the recent publication of the specification. It was felt that the ODNs should ‘own’ the service specification at a local level and take the initiative to reach out to providers. Communication with designated hospitals was identified as an area for improvement, which could be solved by better relationships with local networks. Suggested initiatives included an annual national AYA Cancer Service Away Day, ‘Service Specification Days’ hosted by ODNs with staff from local providers, and easy-to-read service specification guidance documents for different groups within the NHS. Where possible, participants felt there should be a clear stance in the language and standards contained within the service specification in order to avoid variation in implementation between ODNs.

#### 3.2.2. Harnessing the Ideas of Young People

All participants were in favour of involving young people with experience of cancer in the design of service provision, but there was no agreement on how this could be achieved nationally. The use of online platforms such as Microsoft Teams and online surveys could engage groups of young people at a local level. Participants acknowledged that any young person representative forum would need to account for the diversity of young people’s backgrounds and experiences, as well as different ages and tumour types. A suggestion was that this was a critical role of the third sector, to support the voice of young people.

#### 3.2.3. Evaluation of Outcomes and Establishing a National Dashboard

It was suggested that an expert working group could be established to identify key metrics beyond those listed in the service specification. Before any dashboard could be introduced, sufficient notice would be needed to allow teams to ensure any data they submitted was of sufficient quality. There was some discussion of whether dashboards should be publicly accessible or if access should be limited (whether to some data or to some stakeholders). Those responsible for providing data needed to see the utility and value in providing data to ensure quality, reliability, and timeliness. All agreed that dashboards should have utility for clinicians; any dashboard should be user-friendly and created with the clinician’s care delivery needs in mind. Finally, providers should also have the ability to have local metrics, and the dashboard should be flexible in order to accommodate this.

### 3.3. Recommendations from Young People

One of the recommendations from stakeholders during the Policy Lab was for information in a form that would facilitate sharing the key messages to professionals in the designated hospitals. This would raise the importance of viewing the needs of AYAs differently, increase their awareness of the service specification, and highlight the expectation of their roles and subsequent actions if they were treating a young person. Consequently, we held a workshop with the YAP (one had joined the Policy Lab). The purpose was to identify the content of an animation.

After presenting the key changes proposed in the service specification, we asked the YAP to discuss what was important for the adult DH hospital to know. Through discussion, the YAP identified four key messages:


*“The most important thing to remember is we’re a young person first, cancer patient second”*


Provide a named key worker, one person who can advocate for young people across different departments and hospitals where care may be being delivered.More information about clinical trials to understand why they are being done.Please ensure my tumour is banked.When treatment ends, have someone in the team available so we can still contact you and provide information on how we can do this.

The final version of the animation was circulated to the ODNs with the Policy Lab report (Appendix A; https://www.youtube.com/watch?v=go0DogjGUmw (accessed on 23 May 2025)).

## 4. Discussion

BRIGHTLIGHT was a large, complex, mixed-methods, multi-study evaluation of AYA cancer services in England that generated divergent results. Further, some of these results were in conflict with the recommendations set out in the service specification for the commissioning of services for AYAs. We have described a multi-stakeholder approach, including patients, to generate guidance that combines the current research evidence with policy.

The Policy Lab methodology that was utilised represents an effective method for bridging policy–evidence gaps. Policy Labs bring together researchers, practitioners, policymakers, and other stakeholders (including service users) to explore a specific policy challenge in a tight time-frame—the specificity and focus sets this approach apart from other tools that seek to build consensus. The Lab was conceived with a specific key question in mind (“What is a roadmap for empowering different stakeholders to shape how the teenage and young adult service specifications are implemented?”). Facilitation for the Lab sought to ensure that discussions remained focused around this key question, ultimately ensuring that by the conclusion of the day, participants had created a number of feasible and potentially impactful policy recommendations. The recommendations below were co-produced by Lab participants, including practitioners, policymakers, and young people with lived experience of a cancer diagnosis, and succeeded in gaining buy-in from these disparate groups. Focused and skilled facilitation is required in order to ensure that agreement is reached by the conclusion of the workshop, and follow-up is required in order to maintain interest in the recommendations from Lab participants. As described above, the Lab was followed up with a shorter online workshop and semi-structured interview that sought to clarify conclusions and maximise the potential impact from the findings.

The first recommendation by stakeholders was to launch the specification with ‘compelling communication’, maximising awareness, but this was not realised. The service specification was released without much public or healthcare professional communication. The reasons for this were unclear; however, as its release was known to be imminent, the ODN leads were poised for its release and ensured it was accessed and communicated rapidly across the ODNs, and immediate actions were undertaken to start implementing it. As part of our multi-stakeholder approach, we worked with young people to develop promotional materials for the PTC to use across the ODN to raise the profile of the service specification profile in the designated hospitals.

The second recommendation was to harness the ideas of young people nationally. The concept of patients being at the heart of the NHS and research is not new in the UK, and similar philosophies exist in other AYA cancer services with similar age ranges, such as Australia (https://www.canteen.org.au/about-us/youth-leadership (accessed on 23 May 2025)). In particular, AYA cancer care in the UK has a philosophy of the ‘patient’, ‘consumer’, or ‘patient advocacy’ at the centre of any service or research design. Young people are represented or directly involved along the care and research pathway. For example, represented on the Clinical Reference Group, in the design of specialist units, working with national researchers and research groups, and setting the national research priorities. It is unsurprising, then, to have ‘harnessing ideas of young people’ as a recommendation. How this can be done nationally in a cohesive, coordinated, and sustainable manner needs some consideration to ensure inclusivity and representation. Involving young people with cancer brings with it a diverse age range, numerous cancer types, life-stage commitments (at school, college, university, apprentice, graduate roles, and those who are already parents, as some examples), ethnicity, gender and sexuality, as well as geographical and socioeconomic variances across the UK. It is likely that a national approach will require a coordinated approach from third-sector organisations and local ODNs. AYA services in Australia also encompass AYAs diagnosed at 15–24, similar to the UK, and also adopt a patient-centric approach to service design and delivery.

Finally, the third recommendation was to evaluate AYA patient outcomes/experiences and establish a national dashboard of AYA ODN performance. Patient experience is central to healthcare service and delivery in England, and capturing metrics to measure performance has become standard within cancer. For example, the National Cancer Patient Experience Survey has been conducted annually since 2010, which has identified important changes in practice that can improve patient experience [34,38], and quality of life has also been measured through the NHS England Cancer Quality of Life Survey since 2020, which has shown that quality of life is lower in people treated for cancer than the general population [39,40]. However, it is not possible to make inferences from these results to young people because 0.2% of responses in the National Cancer Patient Experience Survey in 2022 were from 16- to 24-year-olds, and the Cancer Quality of Life Survey categorises all patients under 50 as one group.

Currently, in England, an AYA-specific holistic needs assessment is being used, so there is the potential for adding outcome and experience to this. Two key issues will need to be addressed to ensure this provides clinically useful results: the outcomes measured need to reflect issues pertinent to young people; and the disease and treatment questions need to reflect current treatments. Many outcome measures were developed over 20 years ago, so they may not capture the impact of current treatments, especially for AYAs [41]. While the BRIGHTLIGHT cohort study indicated no quality of life, patient-reported outcome, or survival benefits for care within the PTC [24,26,30], the free-text comments captured alongside the standardised instruments were more positive [30]. Potentially, the outcomes and/or the measures used to capture these were not suitable for identifying outcomes associated with specialist care for AYA. There is a compelling reason to identify and measure outcomes associated with specialist AYA care due to various models of the delivery of AYA cancer care being implemented internationally [42,43], but the benefits of these services have not been quantified.

The BRIGHTLIGHT study results and service specification recommendations were not in alignment. The gap in translating evidence, research, and knowledge into practice is well known and reported as taking 17 years to inform care [44]. However, the delay in implementing evidence can have consequences for both patients and the service. This was reflected in the current scenario, where the service specification recommended a model of care that was historically associated with poorer outcome at a greater cost to the NHS and patients [1]. Evaluating healthcare is recognised as being challenging because of its complex and dynamic nature; in recognition of this, BRIGHTLIGHT was designed to look at the environment of care, the workforce, and young people/caregivers’ experience and outcomes. This allowed us to examine multiple data sources with BRIGHTLIGHT and to provide explanations for the cohort results: that it was dependent on an AYA culture for coordinated care, which takes time to develop, and that it was not established nationally when the BRIGHTLIGHT cohort was recruited [1]. This is currently being investigated, but until completed, the potential impact of ‘joint’ care remains unknown.

The current study had a number of limitations. Participants in the Policy Lab were predominantly AYA professionals, i.e., had already bought into the concept of AYA-specific cancer care and the unique needs of AYAs; therefore, the views of those who do not realise the benefit of additional holistic support were not captured. In a bid to address this, the second workshop was conducted to try and obtain input from professionals in the designated hospitals, but the workshop coincided with industrial action by junior doctors, so only 50% of those registered attended. Second, young person representation was through the BRIGHTLIGHT YAP, who are knowledgeable about cancer services and research but were treated some time ago. Their opinions may, therefore, not reflect those of young people who have been treated recently. Despite these limitations, there was wide representation of professional groups from across all thirteen PTCs. Coincidentally, it was also timely and aligned with the release of the service specification. At the time of the Policy Lab, the recommendations were currently relevant and of use to the ODNs. Outputs from this Policy Lab have fed into the proposed new 10-year Cancer Plan Children and Young Person Specific recommendations for England.

## 5. Conclusions

Policy and research are not always aligned, and when emerging evidence does not support current guidance, further exploration is required. Through involving a wider group of clinicians in collaboration with young people, it was possible to gain an alternative perception based on current knowledge and context. This additional insight enabled practical recommendations to be identified to support bridging the gap between BRIGHTLIGHT results and the implementation of the service specification.

## Figures and Tables

**Figure 1 healthcare-13-01821-f001:**
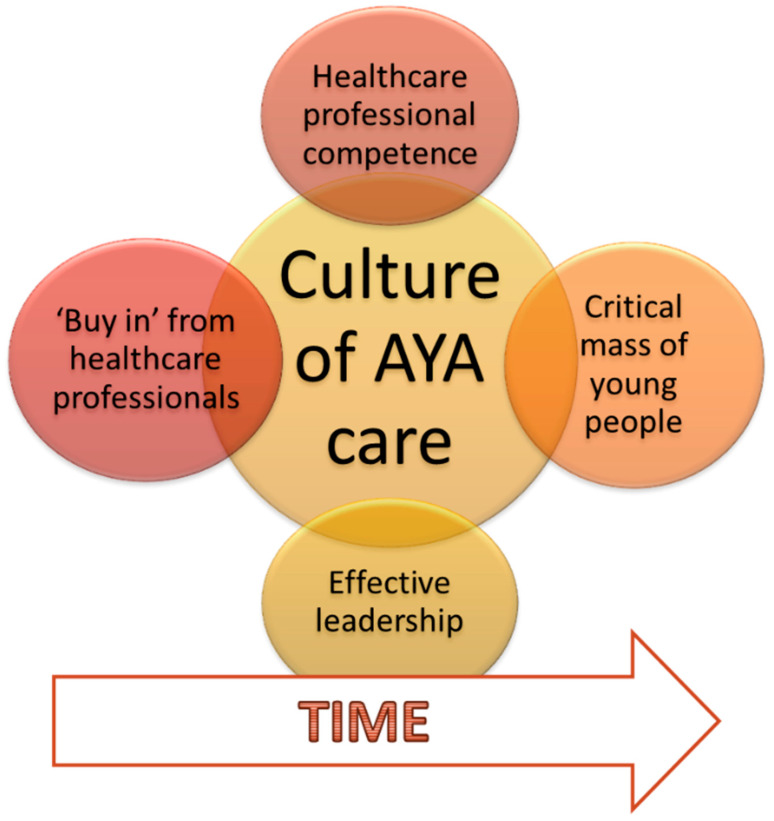
The unique culture of AYA care.

**Figure 2 healthcare-13-01821-f002:**
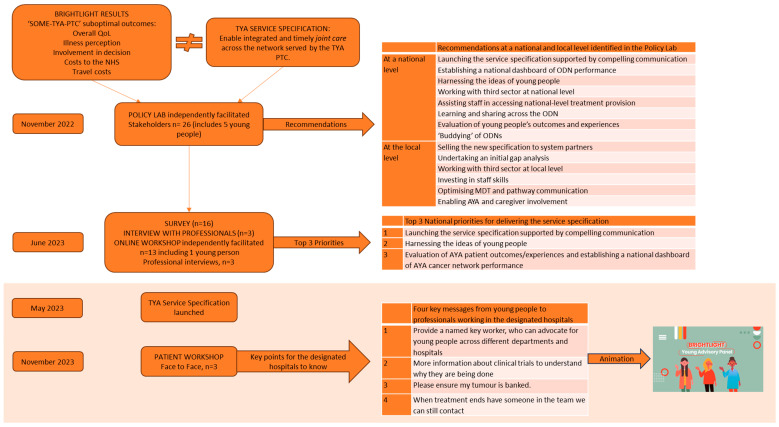
Overview of the methods and results.

**Table 1 healthcare-13-01821-t001:** Summary of the BRIGHTLIGHT cohort findings according to categories of care.

	No Care in the PTC	Some Care in the PTC	All Care in the PTC
Best outcome	QoL at diagnosis Illness perception Costs to the NHS Travel costs Out-of-pocket expenses	Caregiver support, information and services specific for them	Rate of improvement in QoL over time Processes associated with specialist care Caregiver support, information and services specific for them
Middle		Illness perception Out-of-pocket expenses	Costs to the NHS Travel costs
Worst outcome	Rate of improvement in QoL over time Processes associated with specialist care Caregiver support, information and services specific for them	Overall QoL Illness perception Involvement in decisions Costs to the NHS Travel costs	Illness perception Out-of-pocket expenses
All similar (including where differences were not significant)	Overall satisfaction with care Depression and anxiety Social support		

NHS: National Health Service; PTC: Principal Treatment Centre; QoL: quality of life.

**Table 2 healthcare-13-01821-t002:** Top five areas of competence for healthcare professionals caring for young people with cancer.

Rank	Skill	Knowledge	Attitude	Communication
1	Identify the impact of disease on young people’s lives	Know about side effects of treatment and how this might be different to those experienced by children and older adults	Honesty	Listen to young people’s concerns
2	Have excellent clinical skills	Know how to provide age-appropriate care	Friendly and approachable	Talk about difficult issues
3	Work in partnership with young people	Know about current therapies	Be committed to caring for young people with cancer	Speak to young people in terms that are familiar to them while retaining a professional boundary
4	Able to discuss sensitive subjects, e.g., sexual issues	Know about impact of cancer on psychological development	Be respectful	Tell young people about all aspects of their disease
5	Deliver patient-centred care	Developmental issues related to emerging adulthood	Ability to use humour appropriately when interacting with young people	Act as an advocate for young people

**Table 3 healthcare-13-01821-t003:** Recommendations at a national level identified in the Policy Lab.

Recommendation	Supporting Discussion
Launching the service specification supported by compelling communication	Educate relevant stakeholders on the planned service design and rationale for the change. Emphasise the benefit to young people by implementing the specification. Adhering to the specification will remove geographical inequalities.
Establishing a national dashboard of ODN performance	Data collected as part of the metrics to assess the performance of the ODNs should be presented on a dashboard accessible to all ODNs.
Harnessing the ideas of young people	Establish a young person’s cancer service board (like the NHS Youth Forum ^1^). Help recruit young people to support local ODN work. Support creation of training materials for staff. Act as a ‘sounding board’ for new emerging ideas. Provide feedback on the perception of service development.
Working with third sector at national level	Support young people with information, guidance, and practical support Augment the delivery of care, e.g., funding clinical nurse specialists (ensuring the commissioners consider the ‘quid pro quo’, e.g., commitment to sustain clinical nurse specialist posts when funding comes to an end).
Assisting staff in accessing national-level treatment provision	Disseminating information on how resources available in specific centres can be accessed, e.g., proton beam therapy. Scope for engagement between AYA and adult cancer services to identify shared learning.
Learning and sharing across the ODN	Formation of an ‘ODN of ODNs’ to generate shared learning and match skills and knowledge. Agree on consistent standardised approaches on specific issues, e.g., acceptable levels of psychological support. Avoid unnecessary duplication, so there is a central repository of information and resources that are common across the ODNs.
Evaluation of young people’s outcomes and experiences	Future research on outcome and experience needs to track impacts longer than three years. Future research needs to have greater richness than quality of life assessment.
‘Buddying’ of ODNs	Pair ODNs together as informal partners as an alternative to formal ‘peer review’. Enables openness to share the challenges and focuses on learning and helping each other.

^1^ https://www.england.nhs.uk/get-involved/get-involved/how/forums/nhs-youth-forum/ (accessed on 23 May 2025); AYA: adolescent and young adult; MDT: multidisciplinary team; NHS: National Health Service; ODN: Operational Delivery Networks.

**Table 4 healthcare-13-01821-t004:** Recommendations at a local level identified in the Policy Lab.

Recommendation	Supporting Discussion
Selling the new specification to system partners	Involves distilling and articulating the ‘value proposition’—what does the specification offer young people and the participating organisation? Important to ensure the designated hospitals see the benefits to young people that justify the investments and change processes required. Ensure the ODN’s governance structures fit the local situation. Engage with community and primary care through distributing information and holding conferences and roadshows.
Undertaking an initial gap analysis	From the viewpoint of healthcare professionals and from young people’s perspective on the gap between current provision and the specification. Used to create an action plan.
Working with third sector at local level	Assess what is available locally from the NHS and third sector to ensure there is no unnecessary duplication.
Investing in staff skills	Competence is required, not just in cancer-related care but also in young person-related care. Staff training and development is required. Need to build a culture of AYA cancer care that includes the ‘attitude competencies’ (e.g., style of communication) as much as the ‘knowledge competencies’.
Optimising MDT and pathway communication	High-quality communication amongst professionals involved in the treatment, care, and support of young people. Aim to have MDT discussions for more than 75% of young people in the ODN. Optimise the quality of MDT deliberations by implementing learning feedback loops to understand and learn from what works well.
Enabling AYA and caregiver involvement	Implement supportive patient involvement (potentially represented within the ODN).

## Data Availability

All data are reported, with nothing additional to share.

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
