# Peer review of "When Research Evidence and Healthcare Policy Collide: Synergising Results and Policy into BRIGHTLIGHT Guidance to Improve Coordinated Care for Adolescents and Young Adults with Cancer"

_healthcare, 2025, doi:10.3390/healthcare13151821_

Round 1
Reviewer 1 Report
Comments and Suggestions for Authors
Thank you for the opportunity to review the manuscript; ‘When Research Evidence and Healthcare Policy Collide: Synergizing Results and Policy into BRIGHLIGHT Guidance to Improve the Coordinated Care for Adolescents and Young Adults with Cancer’. The manuscript was very well written and described a participatory process (the policy lab) for bridging a gap between adolescent and young adult (AYA) cancer services policy and evidence. My main observation of the manuscript is that is very descriptive, and quite long. In some places it may benefit from removing information not directly related to the policy lab process, and from providing further detail that may be of interest to readers who may wish to implement use a policy lab to bridge other policy-evidence gaps; how do the authors think readers will use their findings (the policy lab outputs and/or the policy lab process itself)? Please find some more specific suggestions below.
1.2 Introduction - some of this section detailing the history of AYA cancer service development in the UK could potentially be removed, or condensed, as it has little bearing on the described study (lines 83 to 108).
2.3.1 Methods: Summary of evidence used in the policy lab, table 1, table 2, figure 1. Suggest removing these to Appendix 1 as this is simply repeating findings from a previous publication.
3 Results, figure 2 - this was a great help in understanding how the various phases contributed outputs, and who was involved. Suggest making the titles of outputs more descriptive, e.g., Recommendations for...; Top 3 priorities for...; Four key messages for..., to whom.
3.1 stage 7 - the process for reporting the results would probably be better reported in the methods section (how were stakeholders informed of the policy lab outputs?), and the outputs from the policy lab reported in the results section.
Line 347 - remind reader that the participants discussed their ambitions for the service specification in the policy lab (at times it was difficult to follow whether recommendations/priorities/messages came from the policy lab, survey or patient workshop).
Figure 3. The font is too small to read. Suggest landscape orientation on a separate page. In the title (or text), perhaps remind the reader that the discussion topics were drawn from BRIGHTLIGHT's findings about young people's experiences of care (Table 1).
4. Discussion, lines 513-520 - much of this is commentary of what happened after the study - suggest deleting.
4 Discussion - in the third paragraph in which you describe the second recommendation around 'harnessing the ideas of young people', it would be good to discuss this recommendation in relation to the broader literature about how young adult's perspectives have informed the design of specialty cancer services in other jurisdictions.
4. Discussion - I think it would be useful for readers if the policy lab itself was discussed, not just the outputs. For example, what were the perceived benefits of the policy lab over other consensus reaching tools, such as a Delphi process? What were some of the challenges with facilitating the policy lab, how do the authors' experiences compare with other policy labs? If one of the purposes of the manuscript is to demonstrate that a policy lab can be an effective method for bridging policy-evidence gaps, then this information would be important. This goes back to my earlier comments around the purpose of the paper and how the authors see the information being used.
4. Discussion, limitations - a further limitation of the study is that the effectiveness of the road map/recommendation developed by the policy lab haven't been evaluated. Did the policy lab produce recommendations that effectively bridged the policy - evidence gap? Perhaps this is an area for further research?
Author Response
Thank you for the opportunity to review the manuscript; ‘When Research Evidence and Healthcare Policy Collide: Synergizing Results and Policy into BRIGHLIGHT Guidance to Improve the Coordinated Care for Adolescents and Young Adults with Cancer’. The manuscript was very well written and described a participatory process (the policy lab) for bridging a gap between adolescent and young adult (AYA) cancer services policy and evidence. My main observation of the manuscript is that is very descriptive, and quite long. In some places it may benefit from removing information not directly related to the policy lab process, and from providing further detail that may be of interest to readers who may wish to implement use a policy lab to bridge other policy-evidence gaps; how do the authors think readers will use their findings (the policy lab outputs and/or the policy lab process itself)?
- Thank you for your comment and taking the time to review our paper. If we remove the detail from the paper about the content of the policy lab, the reader would not be able to understand where the recommendations come from. The paper that describes the method in detail is provided (Hinrich-Krapels et al), what this paper does, is show the practical application of the method.
Please find some more specific suggestions below.
1.2 Introduction - some of this section detailing the history of AYA cancer service development in the UK could potentially be removed, or condensed, as it has little bearing on the described study (lines 83 to 108).
- We have condensed the text from 249 to 149, please see tracked changed document for details.
2.3.1 Methods: Summary of evidence used in the policy lab, table 1, table 2, figure 1. Suggest removing these to Appendix 1 as this is simply repeating findings from a previous publication.
- We have decided to keep the summary of evidence within the main body of the text, trying to balance reviewers’ comments where we have been asked to provide more information about the BRIGHTLIGHT programme. We feel the tables are a good overview for readers who are not familiar with the BRIGHTLIGHT programme.
3 Results, figure 2 - this was a great help in understanding how the various phases contributed outputs, and who was involved. Suggest making the titles of outputs more descriptive, e.g., Recommendations for...; Top 3 priorities for...; Four key messages for..., to whom.
We have added the following details to the figure: Recommendations at a national and local level identified in the Policy Lab; Top 3 National priorities for delivering the service specification; Four key messages from young people to professionals working in the designated hospitals.
3.1 stage 7 - the process for reporting the results would probably be better reported in the methods section (how were stakeholders informed of the policy lab outputs?), and the outputs from the policy lab reported in the results section.
- The methodology and sequence of reporting is an established Policy Lab Methodology; therefore we cannot change this. However, we have added the following section ”2.8 Dissemination of results to participants”. Participants including the young people were emailed a final report of the Policy Lab. Results were disseminated through normal academic and professional channels. The YAP animation was sent to the young people.
Line 347 - remind reader that the participants discussed their ambitions for the service specification in the policy lab (at times it was difficult to follow whether recommendations/priorities/messages came from the policy lab, survey or patient workshop)
- We have added ‘during the Policy Lab’ where you suggested, line numbers are now not in sequence due to the edits.
Figure 3. The font is too small to read. Suggest landscape orientation on a separate page. In the title (or text), perhaps remind the reader that the discussion topics were drawn from BRIGHTLIGHT's findings about young people's experiences of care (Table 1).
- Changing to landscape does not improve the size of the font. As reviewer 2 has also commented on this figure, we have removed it.
- Discussion, lines 513-520 - much of this is commentary of what happened after the study - suggest deleting.
- We have left this text in as it is critical to the paper, as the No. 1 recommendation was ‘1. Launching the service specification supported by compelling communication’, and therefore, this further illustrates the disconnect between ODNs, clinical services and policy.
4 Discussion - in the third paragraph in which you describe the second recommendation around 'harnessing the ideas of young people', it would be good to discuss this recommendation in relation to the broader literature about how young adult's perspectives have informed the design of specialty cancer services in other jurisdictions.
- Internationally, the age range for TYA/AYA differs ranging from a lower age of 12/13 to an upper range of between 24-39 years. Therefore it is difficult to draw direct comparisons between other jurisdictions. However, Australia are one of the few countries with a similar age range and a patient centric approach to service design and we have added a few sentences to reflect this.
- Discussion - I think it would be useful for readers if the policy lab itself was discussed, not just the outputs. For example, what were the perceived benefits of the policy lab over other consensus reaching tools, such as a Delphi process? What were some of the challenges with facilitating the policy lab, how do the authors' experiences compare with other policy labs? If one of the purposes of the manuscript is to demonstrate that a policy lab can be an effective method for bridging policy-evidence gaps, then this information would be important. This goes back to my earlier comments around the purpose of the paper and how the authors see the information being used.
- We have added into the discussion further information on the benefits of the policy lab recommendations.
- Discussion, limitations - a further limitation of the study is that the effectiveness of the road map/recommendation developed by the policy lab haven't been evaluated. Did the policy lab produce recommendations that effectively bridged the policy - evidence gap? Perhaps this is an area for further research?
- Since the article was written the outputs from this Policy Lab have fed into the proposed new 10-year Cancer Plan Children and Young Person Specific recommendations for England. We have added this to the text.
Reviewer 2 Report
Comments and Suggestions for Authors
This manuscript addresses an important and timely topic. The research is generally well-conducted, the objectives are clearly defined, and the findings provide valuable insights. The manuscript is mostly well-written and structured, with appropriate use of methods and analysis. However, there are a few issues that require minor revisions to enhance the clarity and rigor of the paper. Please explain more about BRIGHTLIGHT Program in the introduction section. In the participant selection please explain about inclusion and exclusion criteria. While the results are clearly presented, the Discussion section would benefit from a slightly deeper interpretation
Author Response
Comments and Suggestions for Authors
This manuscript addresses an important and timely topic. The research is generally well-conducted, the objectives are clearly defined, and the findings provide valuable insights. The manuscript is mostly well-written and structured, with appropriate use of methods and analysis. However, there are a few issues that require minor revisions to enhance the clarity and rigor of the paper.
- Thank you for your comment and taking the time to review our paper.
Please explain more about BRIGHTLIGHT Program in the introduction section.
- We feel there is sufficient information about the BRIGHTLIGHT Programme within the introduction, as there is detailed information in Table 1, Table 2, Figure 1 and ‘Section 2.3.1. Summary of the evidence used in the policy lab’. We have referred the reader to that section in the introduction alerting them to more information to follow. We are also trying to balance reviewer comments about the length of the manuscript. If readers require further information all of the publications are open access.
In the participant selection please explain about inclusion and exclusion criteria.
- We have added the following text. ‘Participant eligibility included all stakeholders involved in the commissioning, provision of and delivery of clinical and psychosocial care of AYA with cancer. ‘Exclusion criteria for professional stakeholders were those not involved in the AYA Cancer’. AYA are eligible to participate in the YAP if they have been diagnosed with any cancer between the ages of 13-24 years and diagnosed within the last 10 years (in keeping with BRIGHTLIGHT recruitment period, 2012-14). Exclusion criteria for the YAP are those with a primary cancer diagnosis aged less than 13 years and older than 25. Please see tracked changes in the text.
While the results are clearly presented, the Discussion section would benefit from a slightly deeper interpretation
- Additional text has been added to the discussion. If this is not at the depth the reviewer requires, if they could be more specific, so we have clear direction on where changes would be beneficial.
Reviewer 3 Report
Comments and Suggestions for Authors
When Research Evidence and Healthcare Policy Collide: Synergizing Results and Policy into BRIGHTLIGHT Guidance to Improve the Coordinated Care for Adolescents and Young Adults with Cancer
- This study analyzed a project developed by the Policy Institute at King's College London, which comprised stakeholders' data to reflect on the BRIGHTLIGHT study and inform the service specification, aiming to improve the coordinated care experience and outcomes for young cancer patients. Some potential weaknesses should be addressed, as stated below:
ABSTRACT:
- The subheading “Background/Objectives” comprises three sentences, the first describing BRIGHTLIGHT, the second declaring the results of BRIGHTLIGHT not being available at the time of the policy development, and the third saying the results were not incorporated into policy. So, the purpose of this article, stating something like “This article aims to do xyz…” is not included in the section dedicated to purpose.
- In the abstract, the study design is unclear.
- Specific methodological details in the methods subsection of the abstract are missing. The authors should describe the nature of data from the Policy Lab, and the analytical methods used (the main section on Methods states Content analysis). The online survey is mentioned but what type data are harvested from the survey what type of analyses were performed is unclear, be they thematic, quantitative, or a combination of both.
INTRODUCTION:
- The introduction section is well written as it adequately describes the healthcare delivery system in UK in general, cancer services delivery to adolescents, and the BRIGHTLIGHT nationally used for evaluation of AYA cancer services in England. The authors should establish the significance and aim of this research article clearly in the final paragraph, starting at Line 181 (and not just that of the project).
METHODS:
- The study design should be clearly described at the beginning of the methods section.
- The number of participants in the study should be clearly stated in the Methods section. In the methods, a mention of 10 youth who were 26-29 at the time of data collection is indicated, yet in the results, “a total of 26 people participated in the Policy Lab” is mentioned. The study participants' numbers and types should be clearly mentioned in the methods section. The sample size or number of study participants is confused by later mention of “Dephi survey.” Since an online survey is mentioned in the abstract as part of the methods, the reader will be confused as to whether that was a part of the previous efforts of data collection for the BRIGHT cohort or for the current study. In conclusion, the reader will benefit from a clear description of how many AYA professionals and young persons participated in the study and at what stage were they recruited.
- Specifics of how the data (qualitative or other forms) were collected, validated, or analyzed are not described. If the Delphi process was a part of this current phase of the research, were the data qualitative (as in its classic form comprising expert opinions and consensus-building), or a combination of quantitative rating scales and/or analysis of agreement yielding statistical measures? The authors should elaborate on this.
- What software were used for the thematic analysis and how were the thematic coding validated is not described by the authors.
- Figure 1: A figure illustrating the conceptual framework for AYA care culture is useful. Authors should ensure that they specify, using arrows, how various elements of care are interlinked with each other.
RESULTS
- Figure 2 presents “an overview of methods and results” in the Results section. The authors have not interpreted/described the results in this figure for the reader. Also, presenting methods in the results is always confusing.
- Figure 3 presents the map of constructs from the discussion of outcome differences. This figure is so cluttered that it is difficult to follow.
DISCUSSION
- The discussion explains the results the results well, but it doesn't do a good job of describing methodological limitations such as those pertaining to the data collection from AYA professionals and youth, the generalization of these findings due to selection bias, and any issues arising from thematic coding in content analysis.
Author Response
When Research Evidence and Healthcare Policy Collide: Synergizing Results and Policy into BRIGHTLIGHT Guidance to Improve the Coordinated Care for Adolescents and Young Adults with Cancer. This study analyzed a project developed by the Policy Institute at King's College London, which comprised stakeholders' data to reflect on the BRIGHTLIGHT study and inform the service specification, aiming to improve the coordinated care experience and outcomes for young cancer patients. Some potential weaknesses should be addressed, as stated below:
- Thank you for taking the time to review our manuscript. We hope we have addressed the weaknesses you have identified.
ABSTRACT:
The subheading “Background/Objectives” comprises three sentences, the first describing BRIGHTLIGHT, the second declaring the results of BRIGHTLIGHT not being available at the time of the policy development, and the third saying the results were not incorporated into policy. So, the purpose of this article, stating something like “This article aims to do xyz…” is not included in the section dedicated to purpose. In the abstract, the study design is unclear. Specific methodological details in the methods subsection of the abstract are missing. The authors should describe the nature of data from the Policy Lab, and the analytical methods used (the main section on Methods states Content analysis). The online survey is mentioned but what type data are harvested from the survey what type of analyses were performed is unclear, be they thematic, quantitative, or a combination of both.
- We have made substantial edits to the abstract encompassing the comments above, thank-you.
INTRODUCTION:
The introduction section is well written as it adequately describes the healthcare delivery system in UK in general, cancer services delivery to adolescents, and the BRIGHTLIGHT nationally used for evaluation of AYA cancer services in England. The authors should establish the significance and aim of this research article clearly in the final paragraph, starting at Line 181 (and not just that of the project).
- We have added the following text to establish the significance (delivery of optimal care for AYA) in the context of a policy recommendation which advocates for a model previously been shown to lead to poorer outcomes and higher costs. ‘We set out to co-develop and deliver a Policy Lab to expedite the implementation of the new service specification in the context of the BRIGHTLIGHT results, examining the roles of multi-stakeholders to ensure service delivery is optimized to benefit AYA patients. Addressing the key question: “What is the roadmap for empowering different stakeholders to shape how the AYA service specifications are implemented?”
METHODS:
The study design should be clearly described at the beginning of the methods section.
- The ‘policy lab’ is the study design, which is specified in the first sentence in the methods. It is a recognised and published methodology and ’Box 1’ immediately after the first paragraph in the method section details the steps. We feel with the additional text added to the abstract and also to the end of the introduction (as per your comments above) then there is sufficient information for the reader.
The number of participants in the study should be clearly stated in the Methods section. In the methods, a mention of 10 youth who were 26-29 at the time of data collection is indicated, yet in the results, “a total of 26 people participated in the Policy Lab” is mentioned. The study participants' numbers and types should be clearly mentioned in the methods section. The sample size or number of study participants is confused by later mention of “Dephi survey.” Since an online survey is mentioned in the abstract as part of the methods, the reader will be confused as to whether that was a part of the previous efforts of data collection for the BRIGHT cohort or for the current study. In conclusion, the reader will benefit from a clear description of how many AYA professionals and young persons participated in the study and at what stage were they recruited.
- We have included additional text about the inclusion and exclusion criteria for participation, based on reviewer 2’s comments. Please see Figure 2 for the detail about how many were included in each workshop or point of data collection.
Specifics of how the data (qualitative or other forms) were collected, validated, or analyzed are not described. If the Delphi process was a part of this current phase of the research, were the data qualitative (as in its classic form comprising expert opinions and consensus-building), or a combination of quantitative rating scales and/or analysis of agreement yielding statistical measures? The authors should elaborate on this.
- The Policy Lab did not include a Delphi Survey; this was part of the original BRIGHTLIGHT Programme. We have added text to support this. ‘The top five healthcare professional competencies identified from the international Delphi survey as part of the BRIGHTLIGHT Programme’. Additional text has been added to section 2.5 to clarify how data were analysed.
What software were used for the thematic analysis and how were the thematic coding validated is not described by the authors.
- We are unclear why the reviewer refers to thematic analysis. The policy lab workshop was analysed using content analysis based on templates used in the workshop. We have added detail about the analysis in section 2.5 but no specific software was used to undertake this. The analysis was reviewed in the pyramid narrative by the research team against the raw data. A sentence has been added to clarify this.
Figure 1: A figure illustrating the conceptual framework for AYA care culture is useful. Authors should ensure that they specify, using arrows, how various elements of care are interlinked with each other.
- They are all interlinked which is why they do not have arrows connecting them. The only arrow needed is the one showing that the service needs time to develop.
RESULTS
Figure 2 presents “an overview of methods and results” in the Results section. The authors have not interpreted/described the results in this figure for the reader. Also, presenting methods in the results is always confusing.
- We have added the following text ‘Figure 2 illustrates an overview of key steps of the Policy Lab, the number of participants and the outputs generated.’ We also feel the diagram adds value with having the processes and how each of the steps informed the outputs is valuable and has been commended by other reviewers, therefore we have left figure 2 positioned at the beginning of the results.
Figure 3 presents the map of constructs from the discussion of outcome differences. This figure is so cluttered that it is difficult to follow.
- We have removed figure based on this and reviewer 1’s comment.
DISCUSSION
The discussion explains the results the results well, but it doesn't do a good job of describing methodological limitations such as those pertaining to the data collection from AYA professionals and youth, the generalization of these findings due to selection bias, and any issues arising from thematic coding in content analysis.
- We have added a paragraph discussing the methodology used. We have already acknowledged the participants as a limitation of this study and as with all studies using a qualitative approach, the aim is not to be generalisable but to generate transferable knowledge.